# Biocides with Controlled Degradation for Environmentally Friendly and Cost-Effective Fecal Sludge Management

**DOI:** 10.3390/biology12010045

**Published:** 2022-12-26

**Authors:** Nataliya Loiko, Oleg Kanunnikov, Ksenia Tereshkina, Timofei Pankratov, Svetlana Belova, Ekaterina Botchkova, Anastasia Vishnyakova, Yuriy Litti

**Affiliations:** 1Winogradsky Institute of Microbiology, Fundamentals of Biotechnology Federal Research Center, Russian Academy of Sciences, 117312 Moscow, Russia; 2Rail Chemical LLC, 105005 Moscow, Russia; 3Semenov Federal Research Center for Chemical Physics, Russian Academy of Sciences, 119991 Moscow, Russia

**Keywords:** biocide, fecal sludge, bronopol, 2,2-dibromo-3-nitrilopropionamide (DBNPA), Sharomix, sodium percarbonate, didecyldimethylammonium chloride, polyhexamethylene guanidine, controlled degradation

## Abstract

**Simple Summary:**

During the storage and/or transportation of fecal sludge (FS), it is treated with biocides, such as quaternary ammonium compounds and biguanide derivatives, to control microbial activity and unpleasant odors. These biocides are very effective, which, however, have drawbacks in the disposal of biocide-treated FS, such as toxicity to activated sludge in biological wastewater treatment plants. In this work, the feasibility of using biocides naturally degradable in alkaline medium for more environmentally friendly management of FS was evaluated. The original strategy was based on alkalinization of the medium due to gradual decomposition of urea in FS. The four selected biocides were shown to effectively control microbial activity and degrade after biocidal function, allowing such biocide-treated FS to be disposed of in wastewater treatment plants in an environmentally sound manner without harming the activated sludge.

**Abstract:**

Didecyldimethylammonium chloride (DDAC) and polyhexamethylene guanidine (PHMG) exhibit high antimicrobial activity and are widely used as biocidal agents in chemical toilet additives for the management of fecal sludge (FS). Disposal of such biocide-treated FS to a wastewater treatment plant (WWTP) is a major environmental problem. It is possible to reduce environmental damage through the use of biocidal agents, which easily decompose after performing their main biocidal functions. In this work, it is proposed to use the fact of a gradual increase in pH of FS from the initial 7.5 to 9.0–10.0 due to the decomposition of urea. Six biocidal compounds were selected that are capable of rapidly degrading in an alkaline environment and one that naturally degrades upon prolonged incubation. Four of them: bronopol (30 mg/L), DBNPA (500 mg/L), Sharomix (500 mg/L), and sodium percarbonate (6000 mg/L) have shown promise for environmentally friendly management of FS. In selected dosage, they successfully reduced microbial activity under both aerobic and anaerobic conditions and are cost-effective. After 10 days of incubation, degradation of the biocide occurred as measured by biological oxygen demand (BOD_5_) in biocide-treated FS. Such FS can be discharged to WWTP without severe damage to the activated sludge process, the need for dilution and additional procedures to neutralize toxicity.

## 1. Introduction

Research related to the management of fecal sludge (FS) is of great importance for urban sanitation, as well as for environmental ecology in general [1,2]. Various measures are taken to limit the indiscriminate discharge of FS in an urban environment or in transport (trains, planes, buses, etc.) [3,4,5]. The most common way to manage FS is to use chemical toilets, where FS is collected in containers and then disposed of centrally [6]. The problem with these toilets is microbial activity in FS, accompanied by foul-smelling gas, especially in warm weather [7,8].

Various physical and chemical methods are used to control the microbial activity, such as UV irradiation [9,10], ozonation [11], chlorination [10,12], use of peracetic acid [13], performic acid [14,15], wood ash [16], hydrogen peroxide [17] and various biocides. The use of biocides is most effective in chemical toilets, as it allows not only to control the activity of microorganisms and the release of unpleasant odors but also to avoid the spread of infections [18,19,20].

Quaternary ammonium compounds (QAC) and polyhexamethylene guanidine (PHMG) are widely used for the management of FS in train lavatories [21]. These substances are biocidal against a number of microorganisms, including fungi, Gram-negative and Gram-positive bacteria, and lipophilic viruses [22]. However, the disposal of FS containing these biocides through discharge to wastewater treatment plants (WWTP) becomes a problem [23]. Many studies have shown that discharge of recalcitrant biocides, such as QAC and PHMG, results in the inhibition of nitrification, denitrification and biochemical oxygen demand (BOD) processes in WWTP [24,25]. One of the ways to reduce the environmental damage to the activated sludge process is the use of more easily degradable biocides, which can degrade after direct use.

Once in toilets, urea slowly decomposes with the release of ammonia, which alkalizes the FS. Due to the high content of urea in FS, the pH can increase from the initial value of 7.5 to 9.0–10.0 [26]. This feature can be beneficial for developing a new way of FS management through the use of biocides with controlled decomposition. It is known that some chemical compounds have a high biocidal efficiency, but at the same time they decompose when the medium is alkalized. We selected six such biocidal substances that are odorless, non-toxic in the applicable concentration ranges, and decompose when the pH rises above 7.0. Their structural formulas are shown in Figure 1.

(1) *Dehydroacetic acid (DA) and its sodium salt (DAN)*. DA, or 3-acetyl-2-hydroxy-6-methyl-4H-pyran-4-one, is a cyclic ketone that has found wide application in food preservation, pharmaceuticals and cosmetics [27,28]. DA is tasteless and odorless and effective under acidic conditions (pH 5.0–7.0 with an optimum of 6.0–6.5) because only the protonated form, which exists at acidic pH, can pass through the cell membrane. Once inside the cytoplasm, DA dissociates, lowers intracellular pH, and disrupts the electrical potential of the microbial cell membranes. Because of its low solubility in water (less than 0.1%), DA is mainly used in the form of a water-alcohol solution, for example, in the preservative «Kem DHA» (Akema Fine Chemicals, Coriano, Italy). [29]. The optimal dosage of the preservative is 0.2–0.8%, in terms of active ingredient DA, 0.016–0.064%. More often, DA is used in the form of sodium salt (DAN), which has higher solubility in water [30,31].

In alkaline conditions, the biocidal activity of DA and DAN decreases, up to complete disappearance, so the use of DA or DAN for the management of FS may be promising.

(2) *Bronopol (B).* Bronopol, 2-bromo-2-nitropropane-1,3-diol, is a white or almost white powder, odorless or with a slight characteristic odor, readily soluble in water [32]. Its aqueous solutions are stable between pH 4.0 and 8.0.

Bronopol has a powerful bactericidal effect [33]. It is effective against Gram-positive and Gram-negative bacteria, some species of algae as well as yeasts and molds [34,35]. Therefore, bronopol is used as a preservative in pharmaceuticals, cosmetics and personal care products individually or in combination with other preservatives at concentrations of 0.01–0.1% [36].

Aqueous preparations are not stable at alkaline pH and are also capable of hydrolysis and photolysis. This feature makes it promising for use in FS management. Natural alkalinization of FS during urea decomposition will eliminate biocide residues and make the waste safe for the activated sludge of wastewater treatment plants.

(3) *2,2-dibromo-3-nitrilopropionamide (DBNPA).* DBNPA is a white crystalline powder, well soluble in water. Aqueous solutions of DBNPA are stable at pH 3.0 to 7.0.

DBNPA is a broad-spectrum biocide [37]. It is used in the pulp and paper industry (for the treatment of paper and cellulose sludge); the oil production industry (biocide for cooling water and water pumped into wells and hydroplastics), and in membrane technologies [38,39,40,41].

DBNPA is readily hydrolyzed under both acidic and alkaline conditions, making it promising for FS management. Above pH 7.5, DBNPA starts to decompose to form less or non-toxic compounds: carbon dioxide, ammonia, bromide ions, dibromacetonitrile and dibromoacetic acid. The approximate half-life of DBNPA is 24 h at pH 7.5, 2 h at pH 8.0, 15 min at pH 9.0. The degradation coefficient depends on pH and temperature [42].

(4) *The preparation Sharomix (SH) is a mixture of isothiazalons.* Isothiazalon (IT) and its derivatives (methylisothiazolinone (MI), methylchloroisothiazolinone (MCI), benzisothiazolinone (BIT), octylisothiazolinone (OIT) and dichloroctylisothiazolinone (DCOIT))—powerful biocides that are used for the preservation of detergents, paints and cosmetics, as well as other everyday products [43]. These compounds are able to diffuse through the bacterial cell membrane and the cell wall of fungi. In the intracellular environment, the electron-deficient sulfur N—S bond of these compounds can react with nucleophilic groups of cellular components, such as cysteine thiols in the active centers of proteins, blocking their enzymatic activity and ultimately causing the death of microbial cells [44,45,46].

Biocidal agents contain both individual isothiazolones and their mixtures. In practice, the biocidal agent SHAROMIX MCI with 1.5% isothiazolones (Methylchloroisothiazolinone 1.1% and Methylisothiazolinone 0.4%) in water is widely used. It is effective in small doses: the minimum inhibitory concentration (MIC) of SHAROMIX MCI against bacteria is 1–3 ppm. The product is stable at pH 4.0 to 8.5; temperature 5 to 50 °C. The biocide decomposition rate increases approximately 10–20 times at 40 °C (compared to 7 °C) and 2000 times when the medium is alkalized to pH 11.0 (compared to 4.5) [43]. This property can be used in FS management.

(5) *Sodium percarbonate (P).* Sodium percarbonate is a crystallosolvate of sodium carbonate and hydrogen peroxide Na_2_CO_3_-1.5H_2_O_2_. It is a white granular substance, easily decomposed by heating, and dissolves quickly in hot water [47]. In cold water, it decomposes very slowly, forming hydrogen peroxide. The end products of sodium percarbonate decomposition in water are water, oxygen and sodium bicarbonate. With the decomposition of sodium percarbonate, the pH of the medium slightly increases [48].

The main use of sodium percarbonate is in the domestic sphere as an oxygenated bleaching component in synthetic detergents and stain removers [49]. Sodium percarbonate is also used in: (1) the chemical industry, as an oxidizing agent [50]; (2) in the textile industry in technological processes of dyeing fabrics and rinsing [51]; and (3) in medicine and other fields, as a means for removing all types of contaminants and disinfecting surfaces and materials [52].

Sodium percarbonate can be used as an environmentally friendly biocidal agent for bio-toilets, as, on the one hand, it has antimicrobial properties and, on the other hand, it decomposes quickly in the aquatic environment.

(6) *Silver citrate (SC).* Silver citrate is a water-soluble salt of silver and citric acid (chemical formula Ag_3_C_6_H_5_O_7_), produced by an electrochemical process and containing 2400 ppm of silver ions [53]. Silver citrate is an effective antibacterial compound widely used in medicine [53,54,55] and veterinary medicine [56]. The recommended concentration of SC is 0.5% and higher [57]. If necessary, SC can be used in combination with other biocides [55].

Silver citrate is sensitive to light and is also unstable at a pH above 7.0 and temperatures above 50 °C. Accordingly, it is expected to decompose when added to FS, the pH of which rises to 9.0–9.7 over time due to the decomposition of urea.

Thus, the above biocidal compounds (Figure 1) are promising for the treatment of FS. Five of them:sodium dehydroacetate (DAN), bronopol (B), 2,2-dibromo-3-nitrilopropionamide (DBNPA), sharomix (Sh), and silver citrate (SC), significantly reduce their activity when the pH of the medium is increased to alkaline values. In the case of sodium percarbonate, the decrease in the biocidal effect is not so much dependent on the pH of the medium as it occurred during incubation due to decomposition with the formation of a mixture of hydrogen peroxide (which eventually decomposes into water and oxygen), Na^+^, and CO_3_^2−^.

The purpose of this work was to assess the feasibility of using these biocides to temporarily inhibit microbial activity in fecal sludge for its subsequent more environmentally friendly disposal. In accordance with the purpose of the work, (1) the biocidal effect of selected biocides were tested against test-microorganisms; (2) the ability to self-degrade after alkali addition to the medium or as a result of FS incubation was evaluated; and (3) the biocidal activity of selected biocides in relation to the FS microbiota was analyzed and the degradation of these biocides after a long incubation (10 days) using the BOD_5_ test was assessed.

## 2. Materials and Methods

### 2.1. Fecal Sludge, Biocides, Microorganisms

Fecal sludge (FS) was sampled at the beginning of June 2019 from the environmentally safe toilet complexes (ESTC) of railway trains of the North-Western branch of the Joint-stock company «Federal Passenger Company» (JSC «FPK», Russia). For this study, FS was sampled from ESTC, that was not supplemented with any biocides.

The following chemicals with antimicrobial properties were used as biocides:Sodium dehydroacetate (DAN, Sigma-Aldrich, Buchs, Switzerland);Bronopol (B, St. Petersburg, Russia);2,2-dibromo-3-nitrilopropionamide (DBNPA, Sigma-Aldrich, Buchs, Switzerland),Sharomix (SH, Ashdod, Israel);Sodium percarbonate (P, Krasnodar, Russia);Silver citrate (SC, Darmstadt, Germany);Biocide “Latrina” produced by Limited Liability Company «Rail Chemical» (Russia). “Latrina” is widely used in the ESTC of JSC «FPK» and by Russian Railways as a toilet chemical additive. The composition of “Latrina” included: didecyldimethylammonium chloride (0.24%) and PHMG (6.5%) (total 6.74%), surfactants (surfactants), perfume and water.

The following test-microorganisms from the collection of Research Center of Biotechnology RAS were used for the determination of minimum inhibitory (MIC) and bactericidal (MBC) concentrations:Gram-negative bacteria *Pseudomonas aeruginosa* 4.8.1 and *Alcaligenes faecalis* DOS7 [58];Gram-positive non-sporulating bacteria *Staphylococcus aureus* 209P and *Micrococcus luteus* NCIMB 13267;Gram-positive spore-forming bacteria *Bacillus subtilis* 534, yeast (eukaryotes) *Yarrowia lipolytica* 367-2.

### 2.2. Cultivation of Test-Microorganisms

The test-microorganisms were grown in lysogeny-broth medium (LB) (Broth, Miller, VWR Life Science, Radnor, PA, United States). Cultivation was carried out in 250-mL flasks with 50 mL of nutrient medium with mixing on an orbital shaker (120 rpm) for 24 h (to the stationary phase) at a temperature of 28 °C. Inoculum (culture at the beginning of the stationary growth phase) was introduced in an amount of 0.25 mL per 50 mL of medium (0.5% vol.).

### 2.3. Determination of MIC and MBC

Biocide aliquots at various final concentrations were added to 25 mL glass test tubes with cotton stoppers containing 5 mL of LB medium. The tubes were then inoculated with test-microorganisms at stationary growth phase and incubated in a thermostatically controlled shaker (28 °C, 120 rpm). After 2 days of incubation, the growth of microorganisms was assessed visually by the appearance of turbidity. The lowest concentration of the biocide, at which no growth of the test-microorganisms was observed, was taken as the MIC. Additionally, after 2 days of incubation, aliquots were plated on an agar LB nutrient medium from test tubes in which no visual microbial growth was observed. The MBC was taken to be the lowest concentration of the biocide in the test tube, in which no growth of microorganisms was observed in the agar medium.

### 2.4. Evaluation of Changes in the Antimicrobial Properties of Sharomix, Bronopol and DBNPA

Aliquots of biocidal agents were added to 25 mL glass tubes with 5 mL of LB nutrient medium and cotton stoppers to final concentrations in the medium corresponding to MIC or higher. Then, 40 μL of 1 N NaOH was added to one part of the tubes, increasing the pH of the medium to 9.0. In another part of the tubes, the pH was first raised to 9.0 by adding 1 N NaOH, then after 10 min incubation, 40 μL of 7% hydrochloric acid was added to restore the pH to 7.0.

The tubes were then inoculated with test-microorganisms as described above. The tubes were placed on a shaker at 28 °C. After 4 days, the growth of microorganisms was assessed visually by the appearance of turbidity. The lowest concentration of the biocidal agents at which no growth of test cultures was observed was taken as the MIC.

### 2.5. Evaluation of Changes in the Antimicrobial Properties of Sodium Percarbonate during Long-Term Storage

Three batches of 25 mL glass tubes with 5 mL of LB nutrient medium and cotton stoppers were prepared in which sodium percarbonate (in the form of powder) was added to the final concentrations in the medium, corresponding to MIC or higher. The tubes were thoroughly mixed and stored for 10 days at 28 °C. After 5, 8, and 10 days, inoculation with test-microorganisms of one of the batches was performed as described above. After inoculation, the tubes were placed on a shaker at 28 °C, and microbial growth was evaluated after 4 days. The lowest concentration of the sodium percarbonate at which no growth of test cultures was observed was taken as the MIC.

### 2.6. Evaluation of the Effect of Biocides on the Microbial Activity of FS under Aerobic and Anaerobic Conditions

A total of 10 mL of FS and a certain amount of biocide were added to 120 mL glass vials. In the control vials, no biocide was added to the FS. For anaerobic experiments, the vials were purged with argon, then sealed with butyl rubber and aluminum caps. For aerobic experiments, the vials were first sealed with butyl rubber and aluminum caps and then 50 mL of extra air was added to the vials using a syringe to avoid a lack of oxygen for aerobic microorganisms in the FS. Incubation was performed for 10 days (the time during which the biocide must perform its biocidal function in the ESTC, according to the Rail Chemical terms) on a thermostatically controlled shaker at 110 rpm and 28 °C. Under anaerobic conditions, the rate of oxygen consumption was taken as the criterion for the rate of FS biodegradation, and under anaerobic conditions, the rate of carbon dioxide accumulation. The experiments were performed in triplicate.

The specific rate of oxygen consumption by the microbial community of FS (in mM O_2_/(mL FS ∗ day)) was calculated based on the concentration of oxygen in the headspace of the vials at the beginning and after 10 days of incubation, according to Equation (1):(1)$$W=\frac{\left(c_{O2\;init}-c_{O2\;fin}\right)*\left(V_{G}+50\right)}{100*22.4*V_{L}*10}\;,$$
where

*c_O_*_2 *init*_—initial concentration of oxygen in the headspace, %vol.;

*c_O_*_2 *fin*_—final concentration of oxygen in the headspace, %vol.;

*V_G_*—volume of the headspace, mL;

50—volume of the extra air added to headspace, mL;

100—per cent, %;

22.4—molar volume of a gas at standard temperature and pressure, L/mol;

*V_L_*—volume of the liquid phase, mL;

10—incubation time, day.

The specific rate of carbon dioxide production by the microbial community of FS (in mM CO_2_/ (ml FS ∗ day)) was calculated according to Equation (2):(2)$$W=\frac{V_{G}*c_{CO2\;fin}}{100*22.4*V_{L}*10}\;,$$
where

*c_CO_*_2 *fin*_—final concentration of carbon dioxide in the headspace, %vol.;

*V_G_*—volume of the headspace, mL; 

100—per cent, %;

22.4—molar volume of a gas at standard temperature and pressure, L/mol;

*V_L_*—volume of the liquid phase, mL;

10—incubation time, day.

### 2.7. Determination of the Total Number of Colony Forming Units (CFU)

Aliquots of FS pretreated with a biocide after assessing microbial activity under aerobic and anaerobic conditions were plated after appropriate dilution on agar LB nutrient medium. The cultures were incubated at 28 °C for 3 days. The number of CFU in the corresponding dilutions was counted and the cell number (CFU/mL FS) was determined.

### 2.8. Evaluation of the Decrease in Biocidal Activity in FS

A total of 80 mL of the native (non-pretreated) FS and a certain amount of biocide were added to 120 mL glass vials. Control vials did not contain biocides. In anaerobic experiments, the headspace was purged with argon before incubation. In aerobic experiments, vials were closed with cellulose plugs to avoid a lack of oxygen for aerobic microorganisms. Incubation was performed for 10 days on a thermostatically controlled shaker (110 rpm and 30 °C). The degree of degradation of the biocide was then assessed in a five-day biochemical oxygen demand (BOD_5_) test.

### 2.9. Analytical Methods

The pH was determined using an FE20 pH meter equipped with an InLab^®^ microelectrode (both Mettler Toledo, Switzerland). The oxygen and carbon dioxide concentrations in the headspace of the vials were determined by gas chromatograph Crystal 5000.2 (Chromatec, Yoshkar-Ola, Russia) as described earlier [59]. BOD_5_ was determined by the OxiTop respirometric BOD measuring system (WTW, Weilheim in Oberbayern, Germany), according to manufacturer’s recommendations.

### 2.10. Statistical Methods

All experiments were performed in triplicate. Statistical analysis was carried out using standard mathematical methods (Student’s t-test and calculation of the standard deviation) using the Microsoft Excel program. The data group was considered homogeneous if the mean square deviation σ did not exceed 10 percent. The differences between the data groups were considered valid under the probability criterion *p* < 0.05.

## 3. Results

### 3.1. Antimicrobial Properties against Test-Microorganisms

The study of the antimicrobial properties of six biocidal agents was carried out using standard techniques for determining MIC and MBC against different groups of test-microorganisms most commonly found in wastewater [60], as well as in fecal sludge [21]: Gram-negative bacteria *Pseudomonas aeruginosa* and *Alcaligenes faecalis*; Gram-positive non-sporulating bacteria *Staphylococcus aureus* and *Micrococcus luteus*; Gram-positive spore-forming bacteria *Bacillus subtilis*, yeast (eukaryotes) *Yarrowia lipolytica*.

As a result of the experiment, the concentrations of the studied biocidal agents were established that prevent the growth of bacteria and yeast (MIC, Table 1) and cause their death (MIC, Table 2).

The sensitivity of the test-microorganisms to biocides varied. The spore-forming bacterium *B. subtilis*, as expected, turned out to be the most resistant to all the biocides studied in our study. Gram-negative bacteria and yeasts showed the greatest sensitivity to DBNPA, non-sporulating Gram-positive bacteria to sodium percarbonate and bronopol. However, it turned out that in order to suppress microbial growth with silver citrate and especially DAN, it is necessary to use high concentrations (higher than those indicated in other studies), which makes their use in the treatment of FS not cost-effective (Appendix A).

Thus, only four biocides were selected for further studies: bronopol, DBNPA, Sharomix, and sodium percarbonate.

### 3.2. Changes in the Antimicrobial Properties of Bronopol, DBNPA, and Sharomix Depending on Changes in the pH of the Medium

According to the literature data, bronopol, DBNPA and Sharomix should lose their antimicrobial properties when the medium is alkaline, which can be used as a mechanism to control their activity. To confirm this, we determined the MIC of these biocides against test-microorganisms at pH 9.0.

A total of 1 N NaOH was added to the growth medium of microorganisms containing biocides, increasing the pH to 9.0. It has also been found that at pH 9.0 the test-microorganisms grow only more slowly. Thus, the growth time of test-microorganisms in determining the MIC was doubled (up to 4 days).

When the medium was alkalized to pH 9.0, the MIC values of all three selected biocides increased 1.5–4 times (Table 3). This suggests that the change in pH caused partial inactivation (destruction) of the biocide in the medium, which can be used in practice to reduce the biocidal effect before discharging these substances into sewage treatment plants, in case they are used for the preservation of FS.

It is noteworthy that in another series of experiments, when the pH was restored from 9.0 to 7.0 by adding a 7% HCl solution, the result was the opposite. In most cases, a decrease in the MIC values of the biocides was observed, i.e., microbial growth in such media became more sensitive to biocides and was observed only at their lower concentration in the medium than in the control (Table 4). Additional studies are required to explain this effect. It is likely that the addition of alkali (1 N NaOH) followed by neutralization with acid (7% HCl) results in the formation of toxic by-products of biocide degradation with high antimicrobial activity.

### 3.3. Changes in the Antimicrobial Activity of Sodium Percarbonate during Prolonged Incubation

As noted earlier, sodium percarbonate after addition to an aqueous medium (or nutrient microbial medium) begins to decompose to form hydrogen peroxide (H_2_O_2_), and hydroxyl radicals (OH.) provide the biocidal effect. At the same time, due to the gradual decomposition of H_2_O_2_, the antimicrobial effect of P decreases.

In the next series of experiments, the formation of H_2_O_2_ followed by its decomposition was simulated [61]. After sodium percarbonate was added to the tubes with nutrient microbial medium, the tubes were thoroughly shaken and left for incubation for 1–10 days at 25 °C. At certain time intervals, MICs were determined as described previously.

It was found that the biocidal effect of sodium percarbonate persists for 8 days against Gram-positive bacteria *S. aureus* and decreases after 10 days: the MIC value increases by 1.25 times. In experiments with the Gram-negative bacterium *P. aeruginosa,* the MIC increased 1.25-fold on day 8 and was 10,000 mg/L. On day 10, this value did not change. The yeast *Y. lipolytica* became less sensitive to the action of a 5-day incubation of sodium percarbonate: the MIC increased 1.2 times to 12,000 mg/L, and for 10 days of incubation it was equal to 15,000 mg/L, i.e., MIC increased by 1.5 times (Table 5).

Thus, the experiments showed that incubation of sodium percarbonate for 10 days leads to a decrease in its biocidal effect against test-microorganisms by 1.25–1.50 times.

### 3.4. Antimicrobial Effect of Bronopol, DBNPA, Sharomix and Sodium Percarbonate on FS Microbiota during Long-Term Incubation under Aerobic and Anaerobic Conditions

In the next series of experiments, the ability of four selected biocides to exhibit an antimicrobial effect on the FS microbiota was evaluated. FS and biocides at concentrations based on previously determined MICs and MBCs were added to glass vials (Table 6). In addition, to compare the antimicrobial activity, the “Latrina” biocide containing 0.24% didecyldimethylammonium chloride (DDAC) and 6.5% polyhexamethyleneguanidine (PHMG) was used. “Latrina” is currently widely used in ESTC; this biocide is very effective but very persistent due to low degradability, therefore not environmentally friendly [21]. Then, the dynamics of O_2_ consumption (aerobic experiment) or CO_2_ accumulation (anaerobic experiment) were monitored for 10 days. At the end of the experiments, the specific rate of oxygen consumption or carbon dioxide accumulation, as well as the total microbial number were determined.

#### 3.4.1. Biocidal Effect of Bronopol on FS Microbiota

Bronopol was added to the FS at concentrations of 3, 10, 30, and 60 mg/L. Low concentrations (3 and 10 mg/L) slightly reduced microbial activity under aerobic conditions (Figure 2), but were quite effective under anaerobic conditions (Figure 2). According to the bends of the curves of O_2_ consumption (Figure 2) and CO_2_ production (Figure 2), it is possible to judge the dynamics of the ongoing processes. In the aerobic experiment at low concentrations of biocide, the O_2_ consumption curves were similar to those in the control, but the residual O_2_ content in the headspace was higher, which indicates the biocidal effect of bronopol. After 7 days, the curves became flatter, which could indicate a decrease in the toxic effect of the biocide.

By adding 30 mg/L of bronopol, it was possible to completely suppress microbial activity under aerobic conditions within one day (flat curve) and when its content was doubled, within two days. Then, the process of O_2_ consumption in these vials began, but it was significantly lower than in the control. After 9 days, the slope of O_2_ consumption in biocide-treated vials becomes flatter, which, apparently, as mentioned above, is associated with a decrease in the toxic effect of the biocide on the FS microbiota. Specific rates of oxygen consumption over 10 days in 30 and 60 mg/L treatments were 80.2 and 65.5% of the control, respectively (Table 7).

Similar trends were observed in the anaerobic experiment. The CO_2_ production curves of the biocide-treated vials were significantly lower than in the control, and their levels were proportional to the biocide concentration applied (Figure 2). After 7–9 days of incubation, an upward trend in the curves was observed, indicating a decrease in the toxic effect. Bronopol at doses of 30 and 60 mg/L reduced the production of carbon dioxide by almost 5 and 10 times (Table 7).

The number of viable microorganisms (in terms of CFU) also decreased in proportion to bronopol concentration. In 60 mg/L treatment after 10 days under aerobic conditions, no viable microorganisms were observed, and under anaerobic conditions, the cell number was 10^3^ CFU/mL (Table 7).

#### 3.4.2. Biocidal Effect of DBNPA on FS Microbiota

The use of DBNPA at a concentration of 10 mg/L, which was lower than the MIC for test-microorganisms, resulted in insignificant suppression of microbial activity in FS under both aerobic (Figure 3) and anaerobic conditions (Figure 3), reducing the specific (over 10 days) oxygen consumption rate by 6.6% and carbon dioxide production by 3.8% (Table 7). The cell number in this treatment also decreased insignificantly compared to the control (Table 7). Higher biocide concentrations, 30 and 60 mg/L, were more effective: specific oxygen consumption rate in these treatments decreased by 18.5 and 20.3%, and carbon dioxide production by 7.2 and 10.9%, respectively (Table 7). However, only at DBNPA concentrations of 100 mg/L and above, a significant inhibition of aerobic and anaerobic microbial activity was observed for 1–2 days, which was reflected in Figure 3 with flatter sections of curves. The highest tested concentration of biocide, 500 mg/L, reduced the rate of oxygen consumption almost three-fold compared with the control, and carbon dioxide production almost six-fold. At the same time, no viable cells were detected under aerobic conditions after 10 days, and their number under anaerobic conditions was 4.1 × 10^2^ CFU/mL, which was 3.5 orders of magnitude lower than in the control. Furthermore, as in the case of bronopol, after 9 days of incubation in the presence of 10–60 mg DBNPA/L, an upward trend was observed in the curves, which indicates a decrease in the toxic effect.

#### 3.4.3. Biocidal Effect of Sharomix on FS Microbiota

The smallest tested concentration (60 mg/L) of Sharomix was effective under anaerobic conditions, reducing the rate of CO_2_ production three-fold (Figure 4, Table 7), while this dosage was insufficient to inhibit microbial activity under aerobic conditions (Figure 4). Increasing the content of Sharomix in FS to 200, 500 and 1500 mg/L resulted in inhibiting the microbial activity under aerobic conditions for one, two or four days (judged by the slope of the curves, which were almost flat), respectively, and reducing the specific carbon dioxide production rate in the anaerobic experiment by 75–93%. The cell numbers in the higher concentration (200–1500) treatments were 2.8 × 10^4^ CFU/mL or less (Table 7). At the highest concentration of Sharomix used, 1500 mg/L, the rates of oxygen consumption and carbon dioxide excretion were 8 and 15 times lower compared with the control, while viable microorganisms were practically not detected. Judging by the change in the slope angle of the curves, the toxic effect of Sharomix decreased after 9 days, which was more noticeable in anaerobic conditions.

#### 3.4.4. Biocidal Effect of Sodium Percarbonate on FS Microbiota

Sodium percarbonate exhibits excellent biocidal properties under aerobic and especially anaerobic conditions (Figure 5, Table 7). The lowest tested biocide concentration, 1000 mg/L, reduced the specific oxygen consumption rate by 10% and the carbon dioxide production rate by 90%. Three times the biocide content (3000 mg/L) reduced these rates by 25 and 94%, respectively, while the cell number in these treatments decreased by more than 3 orders of magnitude. At the maximum tested concentration (15,000 mg/L), the activity of microbial processes in both aerobic and anaerobic conditions practically ceased, and viable microorganisms were not detected.

It should be noted that at the highest dosage of the biocide (15,000 mg/L) in the first two days of the aerobic process, an additional amount of released oxygen (above the control) was recorded due to the decomposition of sodium percarbonate with the formation of H_2_O_2_, followed by further decomposition to O_2_. According to the change in the slope of the oxygen concentration curves, it could be said that after 7 days there was a decrease in the biocidal effect of sodium percarbonate (Figure 5).

#### 3.4.5. Biocidal Effect of “Latrina” on FS Microbiota

For a more complete assessment of the antimicrobial activity of the studied biocides, comparative experiments were carried out under the same conditions using the biocide “Latrina”. It contains 0.24% DDAC and 6.5% PHMG and is currently used in the management of FS by Rail Chemical LLC. The working concentration of “Latrina” is 700 mg/L [21,23]. Two concentrations of the biocide were used for the study: one used in practice and a double concentration (Table 6). The use of “Latrina” at a concentration of 700 mg/L resulted in a significant inhibition of microbial activity under aerobic conditions for a day, after which oxygen consumption noticeably increased (Figure 6). Doubling the biocide dosage inhibited microbial activity under aerobic conditions for two days. In general, the specific rate of oxygen consumption was 27.7 and 38.6% lower than the control, respectively, in the presence of 700 and 1400 mg “Latrina”/L (Table 7).

Under anaerobic conditions, the addition of “Latrina” to FS at a concentration of 700 mg/L significantly inhibited microbial activity (Figure 6), while the specific rate of carbon dioxide production decreased by 81.5% (Table 1). A two-fold increase in the content of the biocide only slightly improved the biocidal activity against anaerobic microorganisms. No changes in the slope of the CO_2_ concentration curves at the end of incubation were observed, which indicates that the activity of the biocide was retained during 10 days of the experiment.

The cell numbers in the presence of “Latrina” in the aerobic experiment decreased by 1.5–3.5 orders of magnitude compared to the control, and by 1.5–3.5 orders of magnitude under anaerobic conditions. Thus, the active ingredients of “Latrina” (QAC and PHMG) exhibit a more pronounced biocidal effect on the anaerobic part of the microbial community of FS.

### 3.5. Reduction of Biocide Toxicity after 10 Days of Incubation

FS samples after incubation under aerobic and anaerobic conditions (see Section 3.4.1, Section 3.4.2, Section 3.4.3, Section 3.4.4 and Section 3.4.5) were used to assess the degree of reduction in the toxicity of biocides in relation to the FS microbiota in the BOD_5_ test. It is implied that in the control samples, the BOD_5_ values should be maximum due to the lack of inhibition. The same high BOD_5_ value could be observed for FS samples with reduced toxicity due to degradation of the biocide. It should be noted that no “seed” was used for the BOD_5_ test; therefore, the oxygen-consuming activity of the FS native microflora was evaluated. In addition, to create favorable conditions, the pH of the medium in all samples was preliminarily adjusted to neutral values of 7.0 ± 0.2 by adding 7% HCl solution.

It turned out that BOD_5_ values for all samples of biocide-pretreated FS, incubated under anaerobic conditions, were comparable to the control (Table 8), which indicates a decrease in the toxicity of the biocide-pretreated FS under these conditions. In addition, it is possible that this effect is also due to the increased activity of the persistent aerobic microbiota of PS, which resumed its activity after 10 days of incubation under anaerobic conditions. In samples incubated under aerobic conditions, a decrease in BOD_5_ was observed both for dosages of “Latrina” (more than twice), and for high concentrations of bronopol (by 73.6%) and Sharomix (68.4%). Interestingly, a two-fold increase in the accepted (by Rail Chemical LLC) dose of “Latrina” (from 700 to 1400 mg/L) had little effect on BOD_5_.

High BOD_5_ values obtained for FS samples pre-treated with DBNPA, sodium percarbonate, as well as bronopol at a concentration of 30 mg/L and Sharomix at a concentration of 600 mg/L indicate a decrease in the toxicity of these biocides at the applied concentrations after 10 days of incubation. Thus, fecal sludge pretreated with these biocides in the indicated doses can be discharged to WWTP without significant damage to activated sludge processes without diluting them and carrying out additional measures to neutralize toxicity. Taking into account the results obtained, the following biocides and their doses can be recommended for environmentally safe management of fecal sludge: DBNPA—500 mg/L, Sharomix—500 mg/L, bronopol—30 mg/L and sodium percarbonate—6000 mg/L.

## 4. Discussion

Thus, the six biocides studied in this work had a strong antimicrobial effect on test microorganisms. Four substances selected for further studies: DBNPA, bronopol, Sharomix and sodium percarbonate were able to suppress the activity of aerobic and anaerobic microflora of the FS for 10 days, which is required for environmentally safe toilet complexes (ESTC) of Russian railway long-distance trains. The antimicrobial activity of biocides depended on the dosage. The use of the highest concentrations resulted in the death of part of the FS microbiota, as indicated by the absence of viable cells in the pretreated FS. However, lower concentrations were also feasible in suppressing the activity of the FS microbiota. Optimal concentrations of biocides were determined by comparing (1) the rate of decrease in O_2_ consumption under aerobic conditions, (2) the rate of decrease in CO_2_ production under anaerobic conditions, and (3) the number of viable cells under both aerobic and anaerobic conditions, with the “Latrina” biocide, which is currently used in ESTC (Table 7, Figure 7).

Application of “Latrina” at a working concentration of 700 mg/L reduced specific oxygen consumption rate by 27.7%, and at a concentration of 1400 mg/L—by 38.6%. The same or greater decrease in the specific rate of oxygen consumption was observed for new biocides at the following concentrations: DBNPA—200 mg/L; bronopol—30–60 mg/L; Sharomix—500 mg/L and sodium percarbonate—6000 mg/L (Figure 7).

Under anaerobic conditions, to obtain results similar to the action of “Latrina” in reducing the specific rate of carbon dioxide production, DBNPA can be used at a concentration of 500 mg/l; bronopol 30 mg/L; Sharomix 500 mg/L and sodium percarbonate 1000 mg/L (Figure 7).

The antimicrobial activity of the new biocides under aerobic conditions in all tested concentrations was stronger than that of “Latrina”: the cell number in the FS after 10 days of incubation was lower than 3.3 × 10^5^ CFU/mL (Figure 7). Viable microorganisms were not found at all in the presence of the highest concentrations of the new biocides. Under anaerobic conditions, the effect of new biocides on the reduction of cell numbers was also significant, in most cases surpassing the effect of “Latrina” (Figure 7).

Despite their biocidal effectiveness, the active ingredients of “Latrina”, DDAC and PHMG, are very stubborn compounds and slowly decompose in the environment. Once discharged to wastewater treatment plants, they can become a major problem in activated sludge operation. Moreover, undegraded biocides discharged with treated water from WWTP can persist in water bodies for a long time and cause the emergence of biocide-resistant microorganisms, which can induce resistance to many different antimicrobial agents. The environmentally friendly biocides used in this study have unique properties. DBNPA, bronopol and Sharomix begin to decompose when the pH rises above 8.0. Decomposition of sodium percarbonate occurs at any pH over time. The pH of FS after 10 days of incubation has been shown to increase to 8.78–8.90 in control, and to 8.64–9.50 in biocide-treated FS (Table 8). This enhances the decomposition of biocides. Using the BOD_5_ test, it was shown that after 10 days of incubation, the biocides lose their antimicrobial properties against the FS microbiota. The values of BOD_5_, indirectly reflecting the health of the microflora, for the biocide-pretreated FS samples (bronopol 30 mg/L and Sharomix 600 mg/L) were close to control (without the use of biocides). On the contrary, “Latrina” continued to exhibit antimicrobial activity. A decrease in the toxic effect of the studied biocides could also be judged from the change in the slope of the curves reflecting the dynamics of O_2_ consumption in aerobic experiments and CO_2_ production in anaerobic experiments. It was noted that after 7–9 days, the slope of these curves (mainly corresponding to low concentrations of biocides) had an upward trend, reflecting an increase in the rate of O_2_ consumption or CO_2_ release.

To assess the economic feasibility of using the proposed biocides for the treatment of FS and to select the most cost-effective ones, an assessment of the costs of their use was carried out. For this, the cost of a single treatment of one ESTC with a volume of 1000 L was calculated, taking into account the cost of biocides at the moment and their recommended dosages (Appendix A). The calculation did not take into account the cost of water, containers, dyes, flavors, surfactants, etc., which may be needed for the production of a biocidal preparation for its commercial use. Since Sharomix is a 1.5% aqueous solution of two Isothiazolones (Methylchloroisothiazolinone 1.1% and Methylisothiazolinone 0.4%), Appendix A shows the calculations for its concentrate.

Bronopol and Sharomix appeared to be the least expensive, followed by DBNPA and sodium percarbonate. Further studies will be aimed at testing the action of biocides, in particular bronopol, Sharomix and DBNPA, under conditions close to real. It would also be interesting to find new means and methods of conservation of FS that bring the least harm to the environment.

The findings of this work could be of great ecological importance. To the best of our knowledge, this is the first study of its kind. Probably, due to commercial interest, such results may not be published. The works related to the development of methods to reduce the anthropogenic load on the ecology are very important. By using more gentle methods and means, we can maintain balance on our planet. If there is no possibility to refuse biocides, we should choose the ones that bring less harm. In this regard, the use of biocides with controlled degradation and low residual biocidal action for the temporary inhibition of microbial activity in fecal sludge can have great practical appeal and be environmentally friendly.

## 5. Conclusions

Quaternary ammonium compounds and guanidine derivatives are currently used in many chemical toilet additives, such as “Latrina”, to control microbial activity in FS. Despite their biocidal effectiveness, these substances are very persistent and slowly degrade in the environment. Thus, the disposal of such biocide-pretreated FS in WWTPs has the adverse effect of suppressing nitrification, denitrification and BOD processes. In this study, it is proposed to use other biocidal substances that are easily degraded after a certain time. Based on the literature data, five biocidal compounds were chosen as such substances: sodium salt of Dehydroacetic acid (DAN), bronopol, DBNPA, Sharomix, and silver citrate, which can rapidly decompose in an alkaline environment. Another substance, sodium percarbonate, naturally decomposes upon prolonged incubation in a liquid medium. Using MIC and MBC tests, it was shown that these substances have good antimicrobial activity. However, it turned out that it was necessary to use high concentrations of silver citrate and especially DAN, and given the high cost of these substances, they were excluded from further studies.

At pH 9.0, the MIC values of bronopol, DBNPA and Sharomix increased 1.5–4 times. This suggests that the change in pH caused partial inactivation (destruction) of the biocide in the medium, which can be used in practice to reduce the biocidal effect before discharging these substances into WWTPs. This is all the better because as urea contained in FS decomposes, pH rises from the initial 7.5 to 9.0–10.0. Additionally, experiments have shown that incubation of sodium percarbonate for 10 days leads to a decrease in its biocidal effect against test microorganisms by 1.25–1.50 times.

DBNPA, bronopol, Sharomix and sodium percarbonate were able to suppress the activity of the aerobic and anaerobic microflora of the FS for 10 days, which is required for Russian long-distance railway trains. Optimal biocide concentrations were determined based on comparison with “Latrina”: DBNPA—200 mg/L; bronopol—30–60 mg/L; Sharomix—500 mg/L and sodium percarbonate—6000 mg/L. The calculation of the estimate showed that the most cost-effective was the use of bronopol and Sharomix, then DBNPA and sodium percarbonate. Further research will be aimed at a detailed study of the application of the proposed biocides in real conditions. In addition, to reducing harm to nature, it is necessary to continue the search for new methods for controlling microbial activity in FS and similar waste.

## Figures and Tables

**Figure 1 biology-12-00045-f001:**
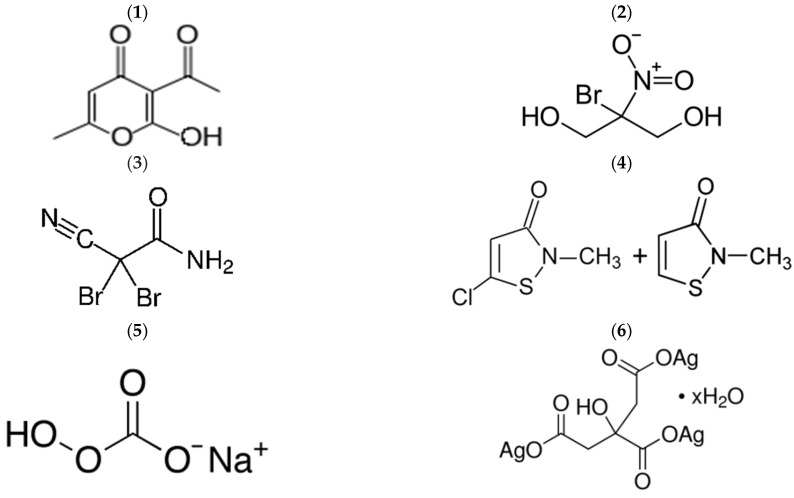
Structural formulas of some of the biocides used in this study. (**1**) Dehydroacetic acid (DA), (**2**) Bronopol (B), (**3**) 2,2-dibromo-3-nitrilopropionamide (DBNPA), (**4**) Sharomix (Methylchloroisothiazolnone + Methylisothiazolinone) (SH), (**5**) Sodium percarbonate (P), (**6**) Silver citrate (SC).

**Figure 2 biology-12-00045-f002:**
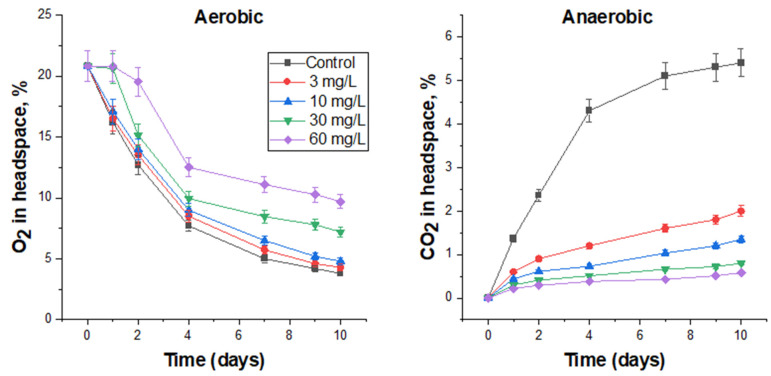
The effect of bronopol on O_2_ consumption under aerobic conditions and CO_2_ production under anaerobic conditions by the respective microbial communities in FS.

**Figure 3 biology-12-00045-f003:**
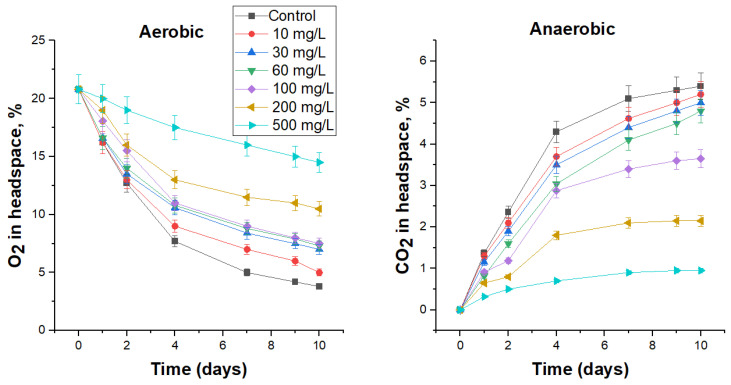
The effect of DBNPA on O_2_ consumption under aerobic conditions and CO_2_ production under anaerobic conditions by the respective microbial communities in FS.

**Figure 4 biology-12-00045-f004:**
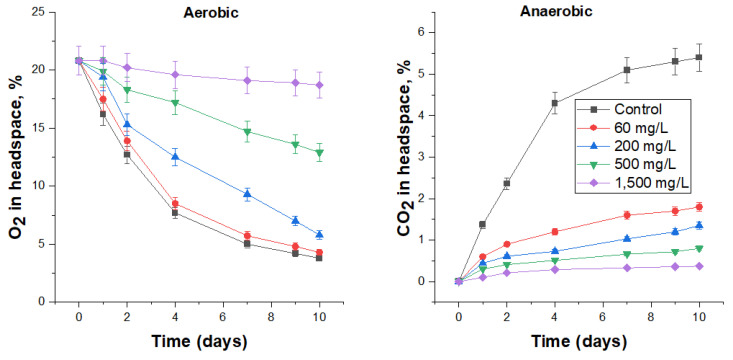
The effect of Sharomix on O_2_ consumption under aerobic conditions and CO_2_ production under anaerobic conditions by the respective microbial communities in FS.

**Figure 5 biology-12-00045-f005:**
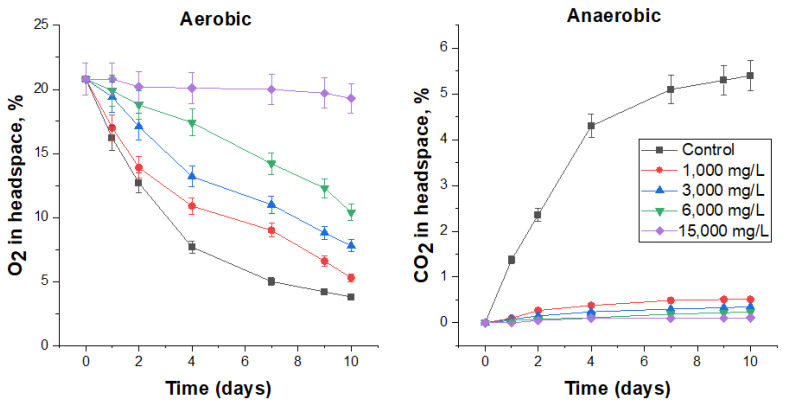
The effect of sodium percarbonate on O_2_ consumption under aerobic conditions and CO_2_ production under anaerobic conditions by the respective microbial communities in FS.

**Figure 6 biology-12-00045-f006:**
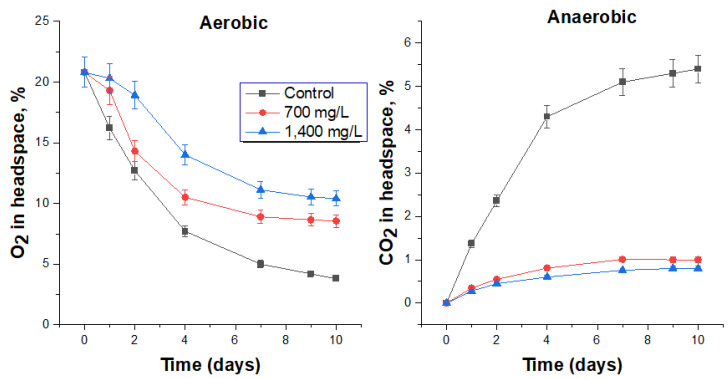
The effect of “Latrina” on O_2_ consumption under aerobic conditions and CO_2_ production under anaerobic conditions by the respective microbial communities in FS.

**Figure 7 biology-12-00045-f007:**
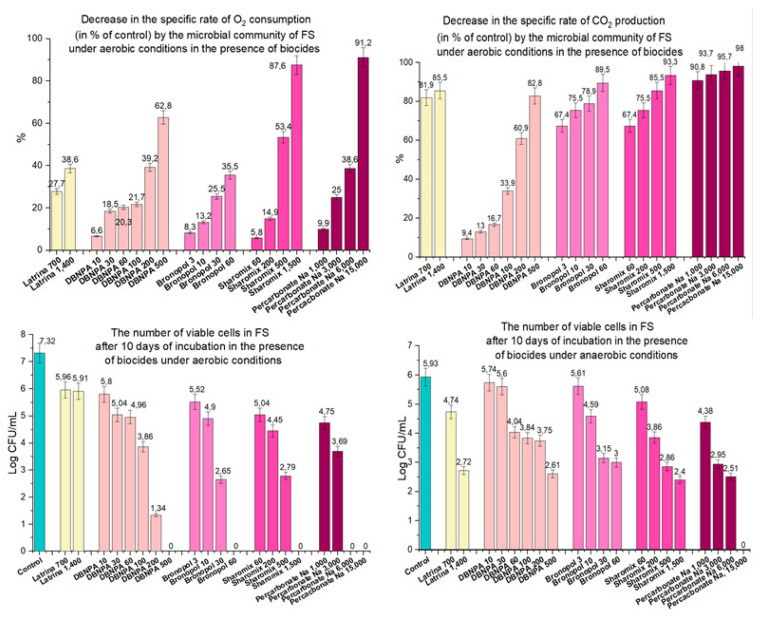
Comparison of biocidal effects of “Latrina” with DBNPA, bronopol, Sharomix and sodium percarbonate.

**Table 1 biology-12-00045-t001:** MIC of the studied biocides for test-microorganisms.

№	Biocides	MIC, mg/L					
		*M. luteus*	*S. aureus*	*P. aeruginosa*	*A. faecalis*	*Y. lipolytica*	*B. subtilis*
**1**	DAN	40,000	40,000	55,000	45,000	20,000	50,000
2	B	30	20	20	40	40	20
3	DBNPA	100	100	25	25	50	200
4	SH	300	100	300	600	300	160
5	P	5000	4000	8000	6000	10,000	10,000
6	SC	20,000	15,000	15,000	15,000	15,000	5000

**Table 2 biology-12-00045-t002:** MBC of the studied biocides for test-microorganisms.

№	Biocide	MBC, mg/L					
		*M. luteus*	*S. aureus*	*P. aeruginosa*	*A. faecalis*	*Y. lipolytica*	*B. subtilis*
**1**	DAN	80,000	90,000	>100,000	>100,000	40,000	>100,000
2	B	40	20	20	40	40	300
3	DBNPA	100	100	25	25	50	800
4	SH	600	300	300	600	300	25,000
5	P	5000	5000	10,000	5000	10,000	40,000
6	SC	40,000	15,000	25,000	25,000	20,000	75,000

**Table 3 biology-12-00045-t003:** MIC of the studied biocides for test-microorganisms at pH 9.0.

Biocide	MIC, mg/L/Change Compared to MIC at pH 7.0, Times		
	*S. aureus*	*P. aeruginosa*	*Y. lipolytica*
B	50/2.5 times higher	40/2.0 times higher	80/2.0 times higher
DBNPA	300/3.0 times higher	100/4.0 times higher	100/2.0 times higher
SH	300/3.0 times higher	600/2.0 times higher	450/1.5 times higher

**Table 4 biology-12-00045-t004:** MIC of the studied biocides for test-microorganisms after the addition of alkali (1 N NaOH) followed by neutralization with acid (7% HCl) from pH 9.0 to 7.0.

Biocide	MIC, mg/L/Change Compared to MIC at pH 7.0, Times		
	*S. aureus*	*P. aeruginosa*	*Y. lipolytica*
B	8/2.5 times less	10/2.0 times less	30/1.3 times less
DBNPA	25/4.0 times less	10/2.5 times less	50/no change
SH	75/1.3 times less	200/1.5 times less	300/no change

**Table 5 biology-12-00045-t005:** Changes in the MIC of sodium percarbonate for test-microorganisms during long-term incubation.

Incubation Time with the Addition of P, Days	MIC, mg/L/Change Compared to MIC at pH 7.0, Times		
	*S. aureus*	*P. aeruginosa*	*Y. lipolytica*
5	4000/no change	9000/1.13 times higher	12,000/1.20 times higher
8	4000/no change	10,000/1.25 times higher	15,000/1.50 times higher
10	5000/1.25 times higher	10,000/1.25 times higher	15,000/1.50 times higher

**Table 6 biology-12-00045-t006:** Ranges of MIC and MBC values of the studied biocides for test-microorganisms and their working concentrations for the treatment of FS.

Biocide	MIC, mg/L	MBC, mg/L	Tested Concentrations, mg/L
B	20–40	20–300	10; 30; 60; 100; 200; 500
DBNPA	25–200	25–800	3; 10; 30; 60
SH	100–600	300–25,000	50; 200; 600; 1500
P	4000–8000	15,000–75,000	1000; 3000; 6000; 15,000
“Latrina”	≥20	≥25	700, 1400

**Table 7 biology-12-00045-t007:** Indicators of microbial activity in FS in the presence of biocides under aerobic and anaerobic conditions.

Biocide, mg/L	Aerobic Conditions			Anaerobic Conditions		
	Specific Rate of O_2_ Consumption		*CFU*/mL	Specific Rate of CO_2_ Production		*CFU*/mL
	mM O_2_/ (mL FS ∗ Day)	in % of Control		mM CO_2_/ (mL FS ∗ Day)	in % of Control	
**0 (Control)**	**0.0121**	100	2.1 × 10^7^	0.00265	100	8.6 × 10^5^
		**Bronopol**				
3	0.0118	97.5	3.3 × 10^5^	0.000982	37.1	4.1 × 10^5^
10	0.0114	94.2	7.5 × 10^4^	0.000663	25.0	3.9 × 10^4^
30	0.00971	80.2	4.5 × 10^2^	0.000571	21.5	1.4 × 10^3^
60	0.00793	65.5	0	0.000285	10.8	1.0 × 10^3^
**DBNPA**						
10	0.0113	93.4	6.3 × 10^5^	0.00255	96.2	5.5 × 10^5^
30	0.00986	81.5	1.1 × 10^5^	0.00246	92.8	4.0 × 10^5^
60	0.00964	79.7	9.2 × 10^4^	0.00236	89.1	1.1 × 10^4^
100	0.00948	78.3	7.3 × 10^3^	0.00179	67.5	6.9 × 10^3^
200	0.00736	60.8	2.2 × 10^1^	0.00106	40.0	5.6 × 10^3^
500	0.00450	37.2	0	0.000467	17.6	4.1 × 10^2^
**Sharomix**						
60	0.0118	97.5	1.1 × 10^5^	0.000884	33.4	1.2 × 10^5^
200	0.0107	88.4	2.8 × 10^4^	0.000663	25.0	7.3 × 10^3^
500	0.00564	46.6	6.1 × 10^2^	0.000393	14.8	7.2 × 10^2^
1500	0.00150	12.4	0	0.000182	6.9	4.0 × 10^2^
**Sodium percarbonate**						
1000	0.0109	90.1	5.6 × 10^4^	0.000250	9.4	2.4 × 10^4^
3000	0.00928	76.7	4.9 × 10^3^	0.000172	6.5	8.9 × 10^2^
6000	0.00743	61.4	0	0.000118	4.5	3.2 × 10^2^
15,000	0.00107	8.8	0	0.0000540	2.0	0
**“Latrina”**						
700	0.00875	72.3	9.2 × 10^5^	0.000491	18.5	5.5 × 10^4^
1400	0.00743	61.4	8.1 × 10^5^	0.000393	14.8	5.2 × 10^2^

**Table 8 biology-12-00045-t008:** BOD_5_ and pH values for control and biocide-pretreated FS samples incubated for 10 days under aerobic or anaerobic conditions.

Biocide	Concentration, mg/L	Aerobic Conditions		Anaerobic Conditions	
		pH	BOD_5_	pH	BOD_5_
**Control**	0	8.9	5690	8.78	5690
Latrina	700	8.92	2050	8.64	5294
	1400	8.89	2027	8.65	4500
DBNPA	200	8.93	5570	8.66	5513
	500	8.94	5510	8.65	5120
B	30	8.85	3720	8.84	4978
	60	8.79	1500	8.79	4710
SH	600	8.9	4856	8.8	4962
	1500	8.92	1800	8.82	4896
P	10,000	9.5	5800	9.34	5450
	15,000	9.57	4235	9.5	4890

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
