# Peer review of "Biocides with Controlled Degradation for Environmentally Friendly and Cost-Effective Fecal Sludge Management"

_biology, 2022, doi:10.3390/biology12010045_

Round 1

Reviewer 1 Report

The article screened and studied the biocidal agents biocidal agents in chemical toilet additives for the management of fecal sludge, which has certain significance for protecting the environment and mitigating the emergence of microbial resistance. The article has the following problems that need to be improved:

 1. The topic is too long and unclear, so it needs to be condensed.

 2. The introduction section is too expatiatory, so it is suggested to condensed it. The performance of several biocidal agents can be expounded in discussion section rather than in introduction section.

 3. The purpose and significance of this study should be clarified in the introduction section.

 4. In 2.3 section, why should the microorganisms be cultured for 2-4 days after adding biocidal agents when the MIC and MBC of the biocidal agents were determined? As the CLSI stipulates that when testing the MIC of drugs on microorganisms, the microorganisms are required to incubate for 24h after adding drugs, whether the incubation time too long in this study? The same problem occurred in 2.4.

Author Response

The article screened and studied the biocidal agents biocidal agents in chemical toilet additives for the management of fecal sludge, which has certain significance for protecting the environment and mitigating the emergence of microbial resistance. The article has the following problems that need to be improved:

Response: Dear reviewer, thank you for your valuable comments.

  1. The topic is too long and unclear, so it needs to be condensed.

Response: The title was condensed

  1. The introduction section is too expatiatory, so it is suggested to condensed it. The performance of several biocidal agents can be expounded in discussion section rather than in introduction section.

Response: The introduction section has been slightly condensed and some paragraphs have been changed. The discussion of the effectiveness of biocides is left in this section, as we consider it important information that brings the reader up to date on the problem.

  1. The purpose and significance of this study should be clarified in the introduction section.

Response: The purpose and significance of this study are clarified.

  1. In 2.3 section, why should the microorganisms be cultured for 2-4 days after adding biocidal agents when the MIC and MBC of the biocidal agents were determined? As the CLSI stipulates that when testing the MIC of drugs on microorganisms, the microorganisms are required to incubate for 24h after adding drugs, whether the incubation time too long in this study? The same problem occurred in 2.4.

Response: Determination of MICs of antibiotics is carried out at a temperature of 37 °C and most often for gram-negative microorganisms. In this case, the bacteria grow within 24 hours. We carried out our experiments to determine MICs at 28 °C (this temperature was important to us, since it was consistent with other research results) and used cultures of gram-positive bacteria (in addition to gram-negative bacteria) that grow more slowly under these conditions (2 days). Therefore, we extended the duration of the MIC determination to 2 days (it is corrected in the text). When we investigated the effect of biocides after alkalization or alkalization and subsequent acidification of the medium, such conditions were not too optimal for bacteria, so we extended the determination time to 4 days.

Reviewer 2 Report

Dear authors,

I recommend that the abstract be 200 words according to the journal's instructions.

Taking into account the efforts of researchers from all over the world to reduce environmental pollution, for the recovery of waste, including sludge from sewage treatment plants, due to the development of urbanization, the amount of sewage sludge has also increased, therefore finding a method to eliminate pathogenic bacteria from sludge like this so that the sludge can be exploited and no longer stored, the use of biocides with controlled degradation and low residual biocidal action for the temporary inhibition of microbial activity in faecal sludge is an ecological and practical solution regarding sewage sludge.The conclusions are consistent with the evidence and arguments presented in the paper, I recommend that they be improved.

I congratulate you for the research done, for the relevant and original subject.

Section 2.2. and 2.3. I recommend that the sentences be reformulated.

Author Response

Dear authors,

I recommend that the abstract be 200 words according to the journal's instructions.

Taking into account the efforts of researchers from all over the world to reduce environmental pollution, for the recovery of waste, including sludge from sewage treatment plants, due to the development of urbanization, the amount of sewage sludge has also increased, therefore finding a method to eliminate pathogenic bacteria from sludge like this so that the sludge can be exploited and no longer stored, the use of biocides with controlled degradation and low residual biocidal action for the temporary inhibition of microbial activity in faecal sludge is an ecological and practical solution regarding sewage sludge.The conclusions are consistent with the evidence and arguments presented in the paper, I recommend that they be improved.

I congratulate you for the research done, for the relevant and original subject.

Section 2.2. and 2.3. I recommend that the sentences be reformulated.

Response: Dear reviewer, thank you for the high appreciation of our work and your comments.

  1. We condensed the abstract and now it has about 200 words.
  2. The sentences in Sections 2.2. and 2.3 were reformulated

Reviewer 3 Report

This paper is recommended for publication with major revision. The work determines the efficacy of four biocides (Bronopol, DBNPA, Sharomix and Sodium Percarbonate) in biocide-treated fecal sludge. The advantage of these biocides over the currently used biocide, “Latrina” is reduced microbial activity under aerobic and anaerobic conditions while natural degradation under alkaline environments provided by decomposition of urea.  

Title:  Biocides with controlled degradation and low residual biocidal 2 effect for temporary inhibition of microbial activity in fecal 3 sludge, followed by its environmentally safe disposal

The title can be crisper. It is too wordy and yet not descriptive of the study.

Line 46-50: All industries and households face the problem of microbial contamination [1, 2]. 46 Microbial activity leads to the deterioration of industrial equipment due to the formation 47 of persistent biofilms and/or the development of corrosion in pipelines, tanks and reser-48 voirs [3-5]. Thus, during food processing, various microorganisms can adhere and aggre-49 gate on the surface of equipment, resulting in the formation of multispecies biofilms [6]. 50 Biofilms are largely responsible for food spoilage, they are also considered responsible for 51 damage to food processing equipment [1].

What does any of this have to do with fecal sludge? The Introduction needs a structured approach. Please revise the problem statement in the context of fecal sludge management rather than biofilms in industries. 

Line 50-53: Also, microbial activity, accompanied by gas 52 formation, creates problems in the management of fecal sludge (FS) in toilets commonly 53 used on city streets, trains, airplanes, buses, etc. [7, 8].

Why is this a problem?

Line 55-57: To prevent microbial biofouling and improve the microbiological quality of 55 wastewater, various physicochemical disinfection methods are used, such as UV irradia-56 tion [9, 10], ozonation [11], chlorination [10, 12], addition of peracetic acid [13], performic 57 acid [14, 15], wood fly ash [16], hydrogen peroxide [17], etc.

Same comment as above. Does not speak to the problem statement of this study.

Line 58-59 Biocidal agents and prepara-58 tions which have antimicrobial activity including against pathogenic organisms, potential 59 sources of infection, are also used [5, 18-20].

What does this mean? Please paraphrase.

Line 89: for example, in the preservative 89 Kem DHA [29].

What is Kem DHA? Please write the full chemical name.

Line 93: In alkaline conditions, the biocidal activity of DC and DAN decreases, up to complete 93 disappearance, so the use of DC or DAN for the management of FS may be promising.

What is DC? Define the acronym/chemical name when it first appears in text.

Line 162 – 163 The purpose of this study was to assess 162 the feasibility of using these biocides for temporary inhibition of microbial activity in fae-163 cal sludge with their subsequent environmentally safe disposal.

Explain here how you will assess feasibility in this study? Also, what evidence is there that the biocides/toilet cleaners used are not environmentally safe for disposal to WWTP?

Section 2: Materials and Methods

Some of the standardized procedures mentioned in the methods section can be moved to a supplementary or referenced to previous studies.

Line 298-302: As a result of the analysis of literature data and the study of the market for biocidal 298 preparations used in industry, at home or in scientific research, we have identified the 6 299 most promising chemicals for use as biocides for the controlled degradation in FS: sodium 300 dehydroacetate (DAN), bronopol (B), 2,2-dibromo-3-nitrilopropionamide (DBNPA), sha-301 romix (Sh), sodium percarbonate (P) and silver citrate (SC) (Table 1). Their properties are 302 discussed in the "Introduction" section.

These sentences should be removed or at the least paraphrased.

Line 309-312: In accordance with the purpose of our work, this study was (1) to test the biocidal 309 effect of 6 selected biocides against test-microorganisms; (2) to evaluate the ability of 4 310 selected biocides to self-degrade after alkali addition to the medium or as a result of FS incubation; (3) to analyze the biocidal activity of 4 selected biocides in relation to the FS 312 microbiota and to asess their degradation after a long incubation (10 days).

These sentences should be moved to before the methods section.

Line 329-330: However, it turned out that in order to suppress microbial growth with silver citrate and 329 especially DAN, it is necessary to use high concentrations (higher than those indicated in 330 other studies), which makes their use in the treatment of FS not cost-effective.

Why not Sodium Percarbonate? Also, provide cost estimates for readers to come to the same conclusion.

Line 357-359: Additional studies are required to explain 357 this effect. It is likely that the addition of alkali (1N NaOH) followed by neutralization 358 with acid (7% HCl) results in the formation of toxic by-products of biocide degradation 359 with high antimicrobial activity.

Cite/refer to the biocide degradation products and the PourBaix diagram for the biocides. This will help supplement the hypothesis. Also find out other studies that may have seen this result for any of the biocides.

Line 367: Simultaneously, H2O2 begins to decompose into water and 367 oxygen, and the antimicrobial effect of P is reduced.

Technically, the biocidal species is hydroxyl radicals (OH.).     

Line 387-388: In the next series of experiments, the ability of four selected biocides to exhibit an 387 antimicrobial effect on the FS microbiota was evaluated. FS and biocides at concentrations 388 based on previously determined MICs and MBCs were added to glass vials (Table 7).

This methods description, including Table 7 should be in the methods and materials section.

Line 409: After 7 days, the curves became flatter, which could indicate a decrease in the 409 toxic effect of the biocide.

Did the authors test for significance?

Line 411-412: By adding 30 mg/l of bronopol, it was possible to completely suppress microbial ac-411 tivity under aerobic conditions within one day (flat curve), and when its content was dou-412 bled, within two days.

So, bronopol had a lag effect on the microbial activity lasting only 1 day and 2 days for the 30 mg/L and 60 mg/L, respectively. What is the recommended dosage for bronopol for use in toilets?   

Figure 1. The effect of bronopol on (a) O2 consumption under aerobic conditions and (b) CO2 pro-430 duction under anaerobic conditions by the respective microbial communities in FS. The numbers in 431 the figure correspond to the dosage of bronopol (in mg/L).

Need better formatting on the figures. Please use Sigmaplot or similar graphing software.

Section 3.4. Antimicrobial effect of Bronopol, DBNPA, Sharomix and sodium percarbonate on FS 385 microbiota during long-term incubation under aerobic and anaerobic conditions

Since, the results for all 4 biocides have similar trends, the authors can summarize (combine/repurpose) the 10 figures (1-5, a&b) and 5 tables into 4 figures and 1 table. While putting the extra information in the supplementary. This section has a lot of good information but the presentation can be much better.

3.5. Comparison of the biocidal effects

So, this section refers to Figure 11-14, which needs to be changed. But overall, the figures 6-9 are using the same information from Table 8-12 and presenting them in relation with the control or with “Latrina”. Since this information was already presented above, you don’t need a whole section to reiterate the results again. Instead, a discussion of the results with similarities and differences between the different biocides will be much appreciated.

Also, please remake the Figures using a graphing software. Additionally, the captions should mention what the ‘red lines’ indicate.     

Line 591-592: Taking into account the results obtained, the following biocides and their 591 doses can be recommended for environmentally safe management of fecal sludge: 592 DBNPA – 500 mg/L, Sharomix – 500 mg/L, Bronopol – 30 mg/L and Sodium Percarbonate 593 – 6,000 mg/L.

Since this is the most important result out of the entire section, preface it by summarizing the criteria used to come to the recommended dosages. At present the recommendation seems only based on BOD5 values, which is not the case.

4. Discussion

This section needs a lot of work. It currently reads more like a summary/conclusion paragraph. You need to add the limitations of the study and any future works that will be helpful to enhance this study. You can also add a discussion on the cost effectiveness or LCA to show the best out of the four biocides. The recommended dosage discussion mentioned above should also go here.

5. Conclusions

Conclusion section needs to include a summary of the entire paper: What was done, why it was studied and the main results.

Author Response

This paper is recommended for publication with major revision. The work determines the efficacy of four biocides (Bronopol, DBNPA, Sharomix and Sodium Percarbonate) in biocide-treated fecal sludge. The advantage of these biocides over the currently used biocide, “Latrina” is reduced microbial activity under aerobic and anaerobic conditions while natural degradation under alkaline environments provided by decomposition of urea.  

Response:  Dear Reviewer, we thank you for the comprehensive and positive review of our manuscript. The comments were very constructive, and we tried to address all of the concerns. Below are the responses point by point.

  1. Title:  Biocides with controlled degradation and low residual biocidal 2 effect for temporary inhibition of microbial activity in fecal 3 sludge, followed by its environmentally safe disposal

The title can be crisper. It is too wordy and yet not descriptive of the study.

Response:  The title was changed to “Biocides with controlled degradation for environmentally friendly and cost-effective fecal sludge management”

  1. Line 46-50: All industries and households face the problem of microbial contamination [1, 2]. 46 Microbial activity leads to the deterioration of industrial equipment due to the formation 47 of persistent biofilms and/or the development of corrosion in pipelines, tanks and reser-48 voirs [3-5]. Thus, during food processing, various microorganisms can adhere and aggre-49 gate on the surface of equipment, resulting in the formation of multispecies biofilms [6]. 50 Biofilms are largely responsible for food spoilage, they are also considered responsible for 51 damage to food processing equipment [1].

What does any of this have to do with fecal sludge? The Introduction needs a structured approach. Please revise the problem statement in the context of fecal sludge management rather than biofilms in industries. 

Response: This part of Introduction was rewritten.   

  1. Line 50-53: Also, microbial activity, accompanied by gas 52 formation, creates problems in the management of fecal sludge (FS) in toilets commonly 53 used on city streets, trains, airplanes, buses, etc. [7, 8].

Why is this a problem?

Response: This can be a problem in warm weather, which favors the microbial activity in fecal sludge, accompanied by malodorous gassing.

This issue has been added to the Introduction section.

  1. Line 55-57: To prevent microbial biofouling and improve the microbiological quality of 55 wastewater, various physicochemical disinfection methods are used, such as UV irradia-56 tion [9, 10], ozonation [11], chlorination [10, 12], addition of peracetic acid [13], performic 57 acid [14, 15], wood fly ash [16], hydrogen peroxide [17], etc.

Same comment as above. Does not speak to the problem statement of this study.

Response: This part of the introduction has been changed to make it more clear.

  1. Line 58-59 Biocidal agents and prepara-58 tions which have antimicrobial activity including against pathogenic organisms, potential 59 sources of infection, are also used [5, 18-20].

What does this mean? Please paraphrase.

Response: It was paraphrased.

  1. Line 89: for example, in the preservative 89 Kem DHA [29].

What is Kem DHA? Please write the full chemical name.

Response: This is the commercial name of the biocidal preparation; it doesn`t have chemical name. To make it clear, we added quotes and the manufacturer. Reference was also added.  

  1. Line 93: In alkaline conditions, the biocidal activity of DC and DAN decreases, up to complete 93 disappearance, so the use of DC or DAN for the management of FS may be promising.

What is DC? Define the acronym/chemical name when it first appears in text.

Response: Sorry, it was a mistake. We meant Dehydroacetic acid (DA). Fixed.

  1.  Line 162 – 163 The purpose of this study was to assess 162 the feasibility of using these biocides for temporary inhibition of microbial activity in fae-163 cal sludge with their subsequent environmentally safe disposal.

Explain here how you will assess feasibility in this study? Also, what evidence is there that the biocides/toilet cleaners used are not environmentally safe for disposal to WWTP?

Response: In the extended aim of the study, we have added a clarification on how we will evaluate the effect of biocides on the microbiota of the fecal sludge. The Introduction also discusses the toxic effects of biocides used in chemical toilets (with a focus on QAC and guanidine derivatives) on wastewater treatment processes.  

Section 2: Materials and Methods

  1. Some of the standardized procedures mentioned in the methods section can be moved to a supplementary or referenced to previous studies.

Response: Since the methods used in this section are not quite standard, and for a clearer understanding, we left them unchanged.

  1. Line 298-302: As a result of the analysis of literature data and the study of the market for biocidal 298 preparations used in industry, at home or in scientific research, we have identified the 6 299 most promising chemicals for use as biocides for the controlled degradation in FS: sodium 300 dehydroacetate (DAN), bronopol (B), 2,2-dibromo-3-nitrilopropionamide (DBNPA), sha-301 romix (Sh), sodium percarbonate (P) and silver citrate (SC) (Table 1). Their properties are 302 discussed in the "Introduction" section.

These sentences should be removed or at the least paraphrased.

Response: We have rephrased this piece of text and moved it to the Introduction section.

  1. Line 309-312: In accordance with the purpose of our work, this study was (1) to test the biocidal 309 effect of 6 selected biocides against test-microorganisms; (2) to evaluate the ability of 4 310 selected biocides to self-degrade after alkali addition to the medium or as a result of FS incubation; (3) to analyze the biocidal activity of 4 selected biocides in relation to the FS 312 microbiota and to asess their degradation after a long incubation (10 days).

These sentences should be moved to before the methods section.

Response: Done

  1. Line 329-330: However, it turned out that in order to suppress microbial growth with silver citrate and 329 especially DAN, it is necessary to use high concentrations (higher than those indicated in 330 other studies), which makes their use in the treatment of FS not cost-effective.

Why not Sodium Percarbonate? Also, provide cost estimates for readers to come to the same conclusion.

Response: Treatment cost estimates are shown in Table S1. Due to the low cost of sodium percarbonate, its use is quite competitive.

  1. Line 357-359: Additional studies are required to explain 357 this effect.It is likely that the addition of alkali (1N NaOH) followed by neutralization 358 with acid (7% HCl) results in the formation of toxic by-products of biocide degradation 359 with high antimicrobial activity.

Cite/refer to the biocide degradation products and the PourBaix diagram for the biocides. This will help supplement the hypothesis. Also find out other studies that may have seen this result for any of the biocides.

Response: We failed to find similar studies, as well as PourBaix diagrams for the compounds used in this work. However, we will take this into account in future work, thanks!  

  1. Line 367: Simultaneously, H2O2 begins to decompose into water and 367 oxygen, and the antimicrobial effect of P is reduced.

Technically, the biocidal species is hydroxyl radicals (OH.).     

Response: Agreed, corrected in the text.

  1. Line 387-388: In the next series of experiments, the ability of four selected biocides to exhibit an 387 antimicrobial effect on the FS microbiota was evaluated. FS and biocides at concentrations 388 based on previously determined MICs and MBCs were added to glass vials (Table 7).

This methods description, including Table 7 should be in the methods and materials section.

Response: Here we cannot agree with the respected reviewer, since we believe that this section does not refer to materials and methods, but describes the results. If it is removed from the Results into Materials and Methods, it will be less clear.

  1. Line 409: After 7 days, the curves became flatter, which could indicate a decrease in the 409 toxic effect of the biocide.

Did the authors test for significance?

Response: The statistical methods used are given in section 2.10, the test for significance was not used in this work. However, we will take this into account in future work, thanks!  

  1. Line 411-412: By adding 30 mg/l of bronopol, it was possible to completely suppress microbial ac-411 tivity under aerobic conditions within one day (flat curve), and when its content was dou-412 bled, within two days.

So, bronopol had a lag effect on the microbial activity lasting only 1 day and 2 days for the 30 mg/L and 60 mg/L, respectively. What is the recommended dosage for bronopol for use in toilets?   

Response: At doses of 30 and 60 mg/l, bronopol completely inhibited the activity of aerobic microorganisms for one and two days, respectively. Its use in the treatment of FS requires only temporary suppression of microbial activity. According to our research, the effective dose for use in toilets is 30 mg/L.

  1. Figure 1. The effect of bronopol on (a) O2 consumption under aerobic conditions and (b) CO2 pro-430 duction under anaerobic conditions by the respective microbial communities in FS. The numbers in 431 the figure correspond to the dosage of bronopol (in mg/L).

Need better formatting on the figures. Please use Sigmaplot or similar graphing software.

Response: Figures are reformatted in OriginPro, should look better now.

  1. Section 3.4. Antimicrobial effect of Bronopol, DBNPA, Sharomix and sodium percarbonate on FS 385 microbiota during long-term incubation under aerobic and anaerobic conditions

Since, the results for all 4 biocides have similar trends, the authors can summarize (combine/repurpose) the 10 figures (1-5, a&b) and 5 tables into 4 figures and 1 table. While putting the extra information in the supplementary. This section has a lot of good information but the presentation can be much better.

Response: Out of 10 graphs, we made 5, and out of 5 tables, we made 1. In addition, we decided not to move anything to the Supplementary, since we consider this information important.

3.5. Comparison of the biocidal effects

  1. So, this section refers to Figure 11-14, which needs to be changed. But overall, the figures 6-9 are using the same information from Table 8-12 and presenting them in relation with the control or with “Latrina”. Since this information was already presented above, you don’t need a whole section to reiterate the results again. Instead, a discussion of the results with similarities and differences between the different biocides will be much appreciated.

Also, please remake the Figures using a graphing software. Additionally, the captions should mention what the ‘red lines’ indicate.     

Response: We removed this section from the Results. Information from this section has been moved to the Discussion. We reformatted the graphs in OriginPro and placed them on one figure.

  1. Line 591-592: Taking into account the results obtained, the following biocides and their 591 doses can be recommended for environmentally safe management of fecal sludge: 592 DBNPA – 500 mg/L, Sharomix – 500 mg/L, Bronopol – 30 mg/L and Sodium Percarbonate 593 – 6,000 mg/L.

Since this is the most important result out of the entire section, preface it by summarizing the criteria used to come to the recommended dosages. At present the recommendation seems only based on BOD5 values, which is not the case.

Response: We agree with the reviewer, we added criteria that resulted in recommended dosages.

  1. Discussion
  2. This section needs a lot of work. It currently reads more like a summary/conclusion paragraph. You need to add the limitations of the study and any future works that will be helpful to enhance this study. You can also add a discussion on the cost effectiveness or LCA to show the best out of the four biocides. The recommended dosage discussion mentioned above should also go here.

Response: We agree with the reviewer, we have made improvements to this section in accordance with the reviewer's suggestions.

  1. Conclusions
  2. Conclusion section needs to include a summary of the entire paper: What was done, why it was studied and the main results.

Response: The conclusions have been rewritten.

Round 2

Reviewer 3 Report

Thank you for addressing my earlier comments.

 I do have a few more minor changes to report: 

1) Line 340: growth time of test-microorganisms in determining the 340 MIC was doubled (up to 4 days). 

Does doubling the MIC determination time warrant halving the concentration to keep the dosage comparable to previous tests? Maybe,  cite/refer any studies using similar method for comparing pH changes.  

2) Line 369-372: It was found that the biocidal effect of sodium percarbonate persists for 8 days 369 against gram-positive bacteria S. aureus and decreases after 10 days: the MIC value in-370 creases by 1.25 times. In experiments with the gram-negative bacterium P. aeruginosa, the 371 MIC increased 1.25-fold on day 8 and was 1%. On day 10, this value did not change. 

What is the 1%? State clearly.

Also, this effect has been reported before in disinfection studies.  https://aricjournal.biomedcentral.com/articles/10.1186/s13756-018-0447-5 

You should compare your results.

3) Figure 6. Comparison of biocidal effects of “Latrina” with DBNPA, Bronopol, Sharomix and Sodium 590 Percarbonate.

Figure 6, x-axis labels and error bars are not readable. Please choose 2-D graph and prime color schemes.  

4)  Lines 578 - 586 and 592-599 are duplicated.

Author Response

Response: Once again, we are thankful to the reviewer for his comments, which helped to improve this work!

I do have a few more minor changes to report: 

1) Line 340: growth time of test-microorganisms in determining the 340 MIC was doubled (up to 4 days). 

Does doubling the MIC determination time warrant halving the concentration to keep the dosage comparable to previous tests? Maybe,  cite/refer any studies using similar method for comparing pH changes.  

Response: The modification of the method for determining the MIC, with the use of which we determined the change in biocidal activity after alkalization of the medium, was modified by us and then experimentally tested. There are no descriptions of such experiments in the literature. Therefore, unfortunately, we cannot provide a reference to similar papers. When choosing the conditions for setting up this experiment, we noted that microorganisms need additional time to adapt to growth at alkaline pH, since the inoculum was grown at pH 7.0. Therefore, we increased the time for assessing the growth of microorganisms to 4 days. This is explained in the article:

Line 337-340: «1 N NaOH was added to the growth medium of microorganisms containing biocides, increasing the pH to 9.0. It has also been found that at pH 9.0 the test-microorganisms grow only more slowly. Thus, the growth time of test-microorganisms in determining the MIC was doubled (up to 4 days).»

2) Line 369-372: It was found that the biocidal effect of sodium percarbonate persists for 8 days 369 against gram-positive bacteria S. aureus and decreases after 10 days: the MIC value in-370 creases by 1.25 times. In experiments with the gram-negative bacterium P. aeruginosa, the 371 MIC increased 1.25-fold on day 8 and was 1%. On day 10, this value did not change. 

 What is the 1%? State clearly.

Also, this effect has been reported before in disinfection studies.  https://aricjournal.biomedcentral.com/articles/10.1186/s13756-018-0447-5 

You should compare your results.

Response: We have corrected the values given in % to the values in mg/L as shown in Table 6. We have provided a citation/reference to the paper you suggested. However, we cannot compare the results presented in this paper with ours, since this paper deals with microbial biofilms. And, accordingly, the methods used to assess the bactericidal activity of biocides are completely different.

3) Figure 6. Comparison of biocidal effects of “Latrina” with DBNPA, Bronopol, Sharomix and Sodium 590 Percarbonate.

Figure 6, x-axis labels and error bars are not readable. Please choose 2-D graph and prime color schemes.  

Response: Fig. 6 has been reformatted

4)  Lines 578 - 586 and 592-599 are duplicated.

Response: Unfortunately, we couldn’t find the duplicated text. This may have happened due to the use of change tracking... please provide an example of a duplicated text so we can fix it